# Effects of spinal cord stimulation on postural control in Parkinson's disease patients with freezing of gait

Andrea Cristina de Lima-Pardini[1†]*, Daniel Boari Coelho[2,3†]*, Carolina Pinto Souza[4], Carolina Oliveira Souza[4], Maria Gabriela dos Santos Ghilardi[4], Tiago Garcia[4], Mariana Voos[5], Matija Milosevic[6], Clement Hamani[7,8], Luis Augusto Teixeira[2], Erich Talamoni Fonoff[4]*

[1]Centre of Mathematics, Computation and Cognition, Federal University of ABC, São Paulo, Brazil; [2]Human Motor Systems Laboratory, School of Physical Education and Sport, University of São Paulo, São Paulo, Brazil; [3]Biomedical Engineering, Federal University of ABC, São Paulo, Brazil; [4]Department of Neurology, University of São Paulo Medical School, São Paulo, Brazil; [5]Department of Physical Therapy, Speech and Occupational Therapy, School of Medicine, University of São Paulo, São Paulo, Brazil; [6]Department of Life Sciences, Graduate School of Arts and Sciences, University of Tokyo, Tokyo, Japan; [7]Division of Neurosurgery, Sunnybrook Research Institute, Harquail Centre for Neuromodulation, University of Toronto, Toronto, Canada; [8]Division of Neurosurgery, Toronto Western Hospital, University of Toronto, Toronto, Canada

*For correspondence:
andrea.cristina@ufabc.edu.br (ACL-P);
daniel.boari@ufabc.edu.br (DBC);
fonoffet@usp.br (ETF)

†These authors contributed equally to this work

Competing interests: The authors declare that no competing interests exist.

**Abstract** Freezing of gait (FoG) in Parkinson's disease (PD) is an incapacitating transient phenomenon, followed by continuous postural disorders. Spinal cord stimulation (SCS) is a promising intervention for FoG in patients with PD, however, its effects on distinct domains of postural control is not well known. The aim of this study is to assess the effects of SCS on FoG and distinct domains of postural control. Four patients with FoG were implanted with SCS systems in the upper thoracic spine. Anticipatory postural adjustment (APA), reactive postural responses, gait and FoG were biomechanically assessed. In general, the results showed that SCS improved FoG and APA. However, SCS failed to improve reactive postural responses. SCS seems to influence cortical motor circuits, involving the supplementary motor area. On the other hand, reactive posture control to external perturbation that mainly relies on neuronal circuitries involving the brainstem and spinal cord, is less influenced by SCS.
DOI: https://doi.org/10.7554/eLife.37727.001

## Introduction

Postural instability and gait disorders (PIGD) are debilitating phenomena that frequently impair loco-motion and can significantly affect the quality of life in Parkinson's disease (PD) patients (*Perez-Lloret et al., 2014*). PIGD is the most common cause of falls, which are associated with increase in morbimortality in PD (*Bloem et al., 2004*; *Delval et al., 2014*). Freezing of gait (FoG) is described as brief and episodic absence or marked reduction of the anterior progression of the feet, despite the intention to walk (*Giladi and Nieuwboer, 2008*). While taking a step forward in unsupported biped position, the body weight is shifted toward the supporting leg. Normal gait requires an exact coordi-nation of postural adjustment in advance of each step forward, namely anticipatory postural

adjustment (APA) (*de Lima-Pardini et al., 2017*). During imminent FoG episodes, the intention to walk is uncoupled from the triggering of APA, with consequent failure of the forward movement. This often results in knee trembling and failure to initiate gait. In PD, FoG episodes are associated with deficient APA (*Mancini et al., 2016*; *Schlenstedt et al., 2018*). Physiological evidence, functional imaging and clinical-pathological studies suggest that FoG is mainly associated with disorders of frontal cortical regions (e.g. supplementary motor area - SMA) that comprise a known brain circuitry dedicated to APA control (*de Lima-Pardini et al., 2017*; *Jacobs et al., 2009a*; *Bolzoni et al., 2015*).

The occurrence of FoG is highly influenced by attention, environmental triggers and dopaminergic drugs, suggesting the existence of complex mechanisms involving brainstem locomotor networks, the basal ganglia and cortical regions (*Jacobs et al., 2009a*; *Deecke, 1987*; *Coelho and Teixeira, 2017*). On the other hand, automatic balance control seems to be mediated mainly by brainstem circuits, being less sensitive to attentional and environmental disturbances (*Horak and Diener, 1994*).

Since the first experimental report from Fuentes *et al.* (*Fuentes et al., 2009*) showing that SCS could enhance locomotion in PD models in mice, and more recently in primates by possibly shifting the brain pathological oscillatory activity (*Santana et al., 2014*), SCS began to figure as a possible treatment for FoG in PD. Increasing evidence suggests that spinal cord stimulation (SCS) improves treatment-resistant postural and gait disorders in patients with PD (*de Andrade et al., 2016*; *Nishioka and Nakajima, 2015*; *Yadav and Nicolelis, 2017*). In primates, SCS increases neuronal firing of the primary motor cortex, and decreases pathological cortico-striatal synchronous low-frequency waves showing that SCS does influence the oscillatory activity in brain motor circuits (*Fuentes et al., 2009*; *Bentley et al., 2016*). In fact, SCS may disrupt the aberrant inhibition of the globus pallidus internus onto thalamus and SMA (*Brudzynski et al., 1993*; *Takakusaki, 2017*). As part of the circuit that controls APA, the SMA has cortifugal projections to pedunculopontine nuclei (PPN), a region particularly involved in gait initiation (*Takakusaki, 2017*; *Aravamuthan et al., 2009*). Since the activity of SMA, globus pallidus and PPN are impaired in patients with FoG (*Snijders et al., 2016*), SCS would be able to modulate this circuit and improve APA and gait initiation. Recently, our group reported positive effects of high-frequency SCS (300 Hz) on gait, improving the performance in various gait tests, which were reproduced during double-blinded assessments. Also, continuous SCS chronically reduced FoG episodes, improved unified Parkinson's disease rating scale (UPDRS)-III motor scale scores and of self-reported quality of life (*Pinto de Souza et al., 2017*). These results are in line with previous clinical observations and findings recorded in Parkinsonian animal models (*Fuentes et al., 2009*; *Santana et al., 2014*). Overall, these preliminary results (*Pinto de Souza et al., 2017*; *Landi et al., 2013*; *Agari and Date, 2012*) suggest that SCS may be a promising approach to attenuate gait disabilities generated by FoG. However, the mechanisms by which SCS affects FoG in PD patients remain unknown. So far, it has been shown that SCS can improve FoG in advanced stages of PD patients; however, the effects of SCS over APA and reactive balance control measurements in patients with PD with PIGD are still unknown. We hypothesize that SCS improves PIGD in patients with PD by improving balance. To test this hypothesis, we studied APA and reactive control of posture and balance using instrumental analysis.

## Results

During 300 Hz-SCS, all participants showed significant decrease of APA duration compared to OFF (*Figure 1C*) (p=0.042). For the 60 Hz-SCS, while participants 1 and 2 showed shortened APA duration, participants 3 and 4 showed increased APA duration as compared to the OFF condition, failing to show significance (p=0.866). Descriptive values of amplitude (peak) of APA showed increased values in all participants, except for the participant 4 at 60 Hz (*Figure 1A*), without reaching significance compared to OFF condition (p=0.177). The improvement in FoG observed in a preliminary report (*Pinto de Souza et al., 2017*) was reproduced in this trial with patients 2 and 3 showing significant reduction of FoG duration during 300 Hz SCS only (p=0.042). Patients 1 and 4 did not experience any FoG episodes during the biomechanical assessment in the Gait Laboratory in any condition (*Figure 1B*).

Considering the reactive postural control, the results of SCS over the amplitude of $COM_{ap}$ and $COP_{ap}$ did not show significant changes for 60 (p=0.860) and 300 Hz (p=0.220) when compared to

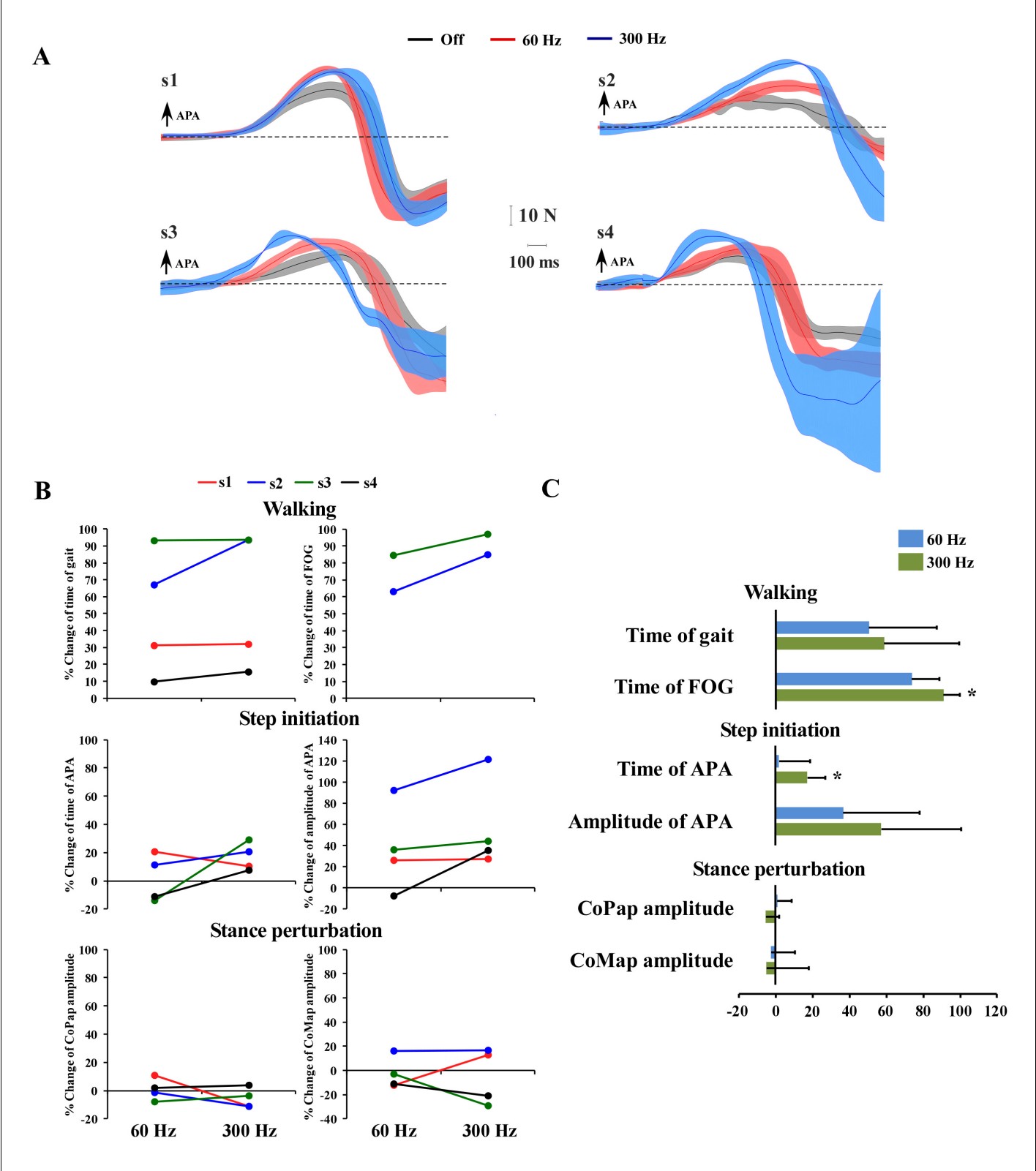

**Figure 1.** Individual and mean values for % change of the outcome variables: time of gait, time of FoG, time of APA, amplitude (peak) of APA and the variables that describe reactive postural control - CoPap amplitude and CoMap amplitude. (**A**) Individual mean curves of the body mediolateral displacement (APA) - thin lines; and the standard deviation (thick and transparent bands) for each stimulation parameter (OFF – gray, 60 Hz – red, 300 Hz – blue). (**B**) Graphs showing the individual mean values of % of change for each outcome variable in 60 and 300 Hz relative to the off

*Figure 1 continued on next page*

*Figure 1 continued*

condition. (C) Mean values and standard deviations of % change for each outcome variable (time of gait, time of FoG, time and amplitude of APA, CoP and CoM amplitude. Asterisks highlight significant effects for 60 and 300 Hz as compared to the off condition.

DOI: https://doi.org/10.7554/eLife.37727.002

OFF state. Note that subject two had higher $COM_{ap}$ (worse performance) during both 60 Hz and 300 Hz SCS, and higher COP (worse performance) during 60 Hz SCS compared to OFF state.

## Discussion

While normal gait requires adequate coordination between the APA and limb movements (*de Lima-Pardini et al., 2017*), FoG episodes are thought to involve defective APA resulting in difficulty or cessation of gait initiation (*Schlenstedt et al., 2018*; *Jacobs et al., 2009b*). In the present study, data showed that SCS at 300 Hz was effective in reducing the duration of APA and reducing the time of FoG. Conversely, we observed that SCS at 60 Hz failed to improve gait and APA. Corroborating clinical results, there was no significant improvement in FoG in the present study, neither in our preliminary report when 60 Hz-SCS was applied (*Pinto de Souza et al., 2017*). Unexpectedly, two patients did not experience any FoG episodes during laboratory examination, even when SCS was OFF for a period longer than their carry-over effect lasted. Being the FoG an episodic phenomenon, even severe patients may not exhibit FoG in the laboratory (*Nutt et al., 2011*; *Giladi et al., 2000*). Conversely, the time of FoG was reduced in the other 2 subjects during 300 Hz SCS. In the previous report, we showed that all subjects did experience frequent FoG episodes and were responsive to 300 Hz SCS in the outpatient clinic (*Pinto de Souza et al., 2017*). Nonetheless, our findings suggest that SCS at 300 Hz is more effective than SCS at 60 Hz (both with pulse width at 90 μs and amplitudes at 105% of sensory threshold) in improving FoG and correcting APAs. However, recent studies have shown that SCS at higher density stimulation parameters (200–500 μs/30–130 Hz) also improved gait and FoG in PD patients (*Samotus et al., 2018*) or at 70–100 Hz in primary freezing of gait when applied at T8-T10 spinal segments (*Rohani et al., 2017*). Yet, none of the reports presented data on SCS at higher frequencies such as 300 Hz. Earlier reports also show that SCS improved gait but these studies did not employ a systematic assessment of gait, posture and FoG, comparing different stimulation frequencies as we showed here (*de Andrade et al., 2016*).

Circuits involving the supplementary motor area (SMA) are thought to modulate APA (*Takakusaki, 2017*), specifically in its timing (*Jacobs et al., 2009a*). Our results showed that timing, but not amplitude of APA was improved by SCS at 300 Hz, which could indicate that motor circuits involving SMA were also influenced. Although data showing that SCS directly modulates SMA still lacks, extensive data show that SCS modulates cortical function of primary sensory and motor areas, in addition to the prefrontal, cingulate and insula cortices, and thalamus in humans and animals (*Fuentes et al., 2009*; *Santana et al., 2014*; *Bentley et al., 2016*; *Stancák et al., 2008*). Inputs from ascending leminiscal and extralemniscal pathways to the brainstem and thalamus that may modulate SMA are also highly connected to the pedunculopontine nucleus (PPN), a locomotor region in the brainstem involved in the control of movement initiation and body equilibrium (*Takakusaki, 2017*). It is thought that ascending stimulation of the thalamic-cortical-striatum loop delivered by SCS may disrupt the aberrant inhibition of the globus pallidus internus onto thalamus and PPN, facilitating the reticulospinal tract and, consequently, activating central pattern generators (CPG) in the spinal cord (*Brudzynski et al., 1993*; *Takakusaki, 2017*). According to previous studies in animals, facilitation of the thalamic-cortical-striatum loop occurs due to the inhibition of abnormal synchronous beta frequency neuronal oscillations (*Fuentes et al., 2009*; *Santana et al., 2014*; *Yadav and Nicolelis, 2017*).

We also observed that the reactive posture control was not affected by SCS in both conditions. In fact, one of the participants showed poorer reactive postural control in both stimulation frequencies, but still exhibiting positive response in APA and FoG. While a case study also found improvements in gait with no changes in reactive postural control after SCS (*Landi et al., 2013*), another report showed an effect of SCS in reactive postural responses (*Agari and Date, 2012*). These studies, however, assessed responses to perturbation by the pull test, which is a qualitative measurement influenced by a subjective impression of the evaluator. Here, we show a more robust and reliable kinetic

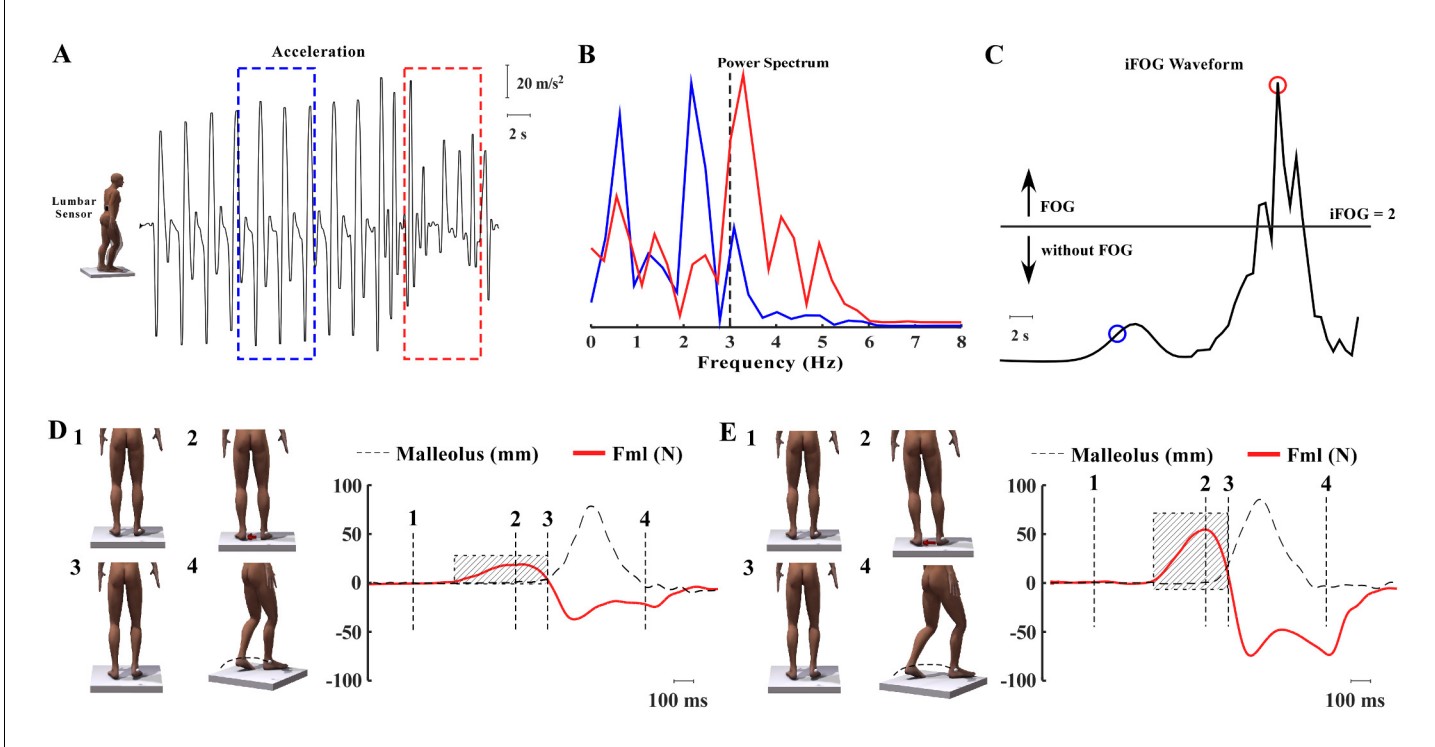

**Figure 2.** Representation of FOG based on spectral analysis and the step initiation task. (A) Characterization of FoG based on spectral analysis. Schematic representation of a man representing the lumbar accelerometer as a small black box. The curve represents the vertical acceleration acquired during step initiation. The dotted square shows a 7.5 s window for the frequency analysis domain (normal gait in blue and freezing in red). (B) Spectral analysis of acceleration showing one band representing the locomotor period (0–3 Hz) and the FoG band (3–8 Hz). (C) FoG index showing the clinical threshold (>2 = FoG); FoG index is calculated by dividing the FoG band by the locomotor band (blue circle – power peak of normal gait; red circle – power peak of a period with FoG). (D) Representation of the step initiation task, showing the marker on the malleolus to detect the moment that the foot clears the floor. The sequence showed in D and E (1-4) shows: (1) beginning of the task, when the participant is in quiet standing (body weight balanced under the feet), preceding the step. (2) The red arrow shows body weight shifting toward the supporting leg (APA). (3) Once the body weight is shifted contralaterally, the moving leg can be released to take a step and finish the movement (4). The graphs showed in D and E represent the moment that each phase of the sequence 1-4) occurs: 1-quiet standing; 2- peak of APA; 3- step; 4-end of the movement. The red line represents the mediolateral force under the supporting leg, the dotted line is the displacement of the moving foot. The red line in D represents a defective mediolateral displacement of the body weight (APA), showing a relative smaller amplitude and longer APA compared to a normal APA displayed in E.
DOI: https://doi.org/10.7554/eLife.37727.003

analysis of postural control. It is worth noting that we employed a systematic method to quantify FoG, considered more reliable than the Freezing of Gait Questionnaire (*Giladi et al., 2000*) alone. Based on the present data and in our previous results, SCS apparently improves FoG and APA and does not influence reactive posture control. It is possible that SCS might differently affect these two mechanisms of postural control – reactive and anticipatory. APA mechanisms are dependent of thalamo-cortical-striatum loops highly influenced by attentional and environmental changes (*Takakusaki, 2017*; *Jacobs and Horak, 2007*). On the other hand, unlike APAs, reactive posture control to external and unpredictable triggers relies on neuronal circuitries involving the brainstem and spinal cord with less cortical influence from the cortex (*Jacobs and Horak, 2007*; *Honeycutt et al., 2009*). It is possible that circuits that depend on cortical influence are more sensitive to SCS, given the increasing in cortical input to the striatum that reaches structures involved in planning of movement, required during APAs but not during unpredictable external perturbations (*Krishnamurthi et al., 2012*; *Bleuse et al., 2011*; *Liu et al., 2006*). Similarly to what have been found for DBS, our results suggest that SCS did not improve reactive postural control (*St George et al., 2012*). Although there is evidence that DBS initially improves FoG and gait (*Schlenstedt et al., 2017*), biomechanical and laboratory findings on APA are still conflicting. A few studies show that DBS impairs APA during step initiation (*Muniz et al., 2010*) even when administered in the ON levodopa state (*Rocchi et al.,*

*2012*). These results suggest that continuous stimulation of circuits in the subthalamic region may lead to plastic changes that impaired step initiation. While DBS corrects cardinal symptoms of PD and dyskinesias, it does not seem to substantially affect APA and FoG. In contrast, SCS can improve FoG and APA in both DBS ON and OFF medication conditions. These collective results suggest that SCS and DBS could be mechanistically and clinically complementary.

In conclusion, our findings suggest that the effects of SCS observed clinically in FOG are followed by an improvement in APA during step initiation, but not by changes in reactive postural control. We note, however, that our findings were recorded in a small number of participants. In the future, following clinical trials with larger sample, the use of similar methods of instrumented posture and gait assessment, may possibly contribute to better understanding the effects of SCS in different gait and postural situations.

## Materials and methods

The study was approved by the department review board and in the ethics committee at Hospital das Clinicas of University of São Paulo and registered in the national clinical research database (CAP-PESQ-HCFMUSP #12690213.0.0000.0068), which requires all participants to be previously instructed about the procedures and to give written informed consent prior to study inclusion. Patients included in the present study participated in the clinical trial also registered in the clinicaltrials.gov (#NCT02388204). All experiments were performed in accordance with the Declaration of Helsinki (*World Medical Association, 2002*). Four patients with PIGD previously treated with bilateral STN DBS were implanted with SCS systems in the upper thoracic spine (T2-T4) (*Pinto de Souza et al., 2017*). Patients were tested for the effects of SCS on APA, and postural reactive responses related to postural and balance control. Assessments were performed in the following conditions: (*Perez-Lloret et al., 2014*) OFF SCS; (*Bloem et al., 2004*) ON SCS at 60 Hz; and (*Delval et al., 2014*) ON SCS at 300 Hz with or without medication (ON/OFF meds) for at least 12 hr. These conditions were randomized and tested at least one week apart. 60 Hz SCS and 300 Hz were applied as active condition since they generated undistinguished paresthesias, but in average 60 Hz exerted no significant effects in gait, while 300 Hz elicited substantial improvement (*Pinto de Souza et al., 2017*).

### Data collection

FoG was assessed using a wireless accelerometer (Delsys, Trigno Delsys Inc., USA; filter: Butterworth, low-pass at 10 Hz) placed on the lumbar region during a 10 m walk task. Three 10 m walking trials were performed. To quantify APA, the patients were next assessed during step initiation with the less affected leg, having both feet positioned on a force platform (AMTI, OR6-6, Advanced Mechanical Technology Inc., USA; with an internal analog low-pass filter with a cutoff frequency set at 100 Hz). A reflective marker on the less affected leg (defined from the UPDRS part III) was used to identify the onset of the step through a motion analysis system (Vicon Nexus V.1.8, UK, with four cameras T10 - RRID:SCR_015001). The APA forces of the more affected side (contralateral limb to the stepping leg) were considered for the analysis. The single step was completed and the moving leg returned to the initial position. To maintain the same feet position over subsequent trials, the original positioning was outlined with tape on the force platform. Participants performed a single block of 10 trials of step initiation, with intertrial intervals of approximately 30 s. They used no physical support or sensory cueing to perform the step. At all times an experimenter remained close by to ensure safety of patients.

We also evaluated reactive postural responses to unanticipated mechanical perturbations. Reactive postural control was assessed using a custom-built movable force platform with an integrated force plate as the base of support (*Coelho and Teixeira, 2017*). Platform displacement was controlled through software developed in LabVIEW (LabVIEW, National Instruments Corp., USA - RRID: SCR_014325). Participants were tested in the upright posture with the feet positioned hip-width apart and slightly oriented sidewards (self-selected as preferred), keeping the arms folded on the chest. The task consisted of recovering stable body equilibrium following unanticipated backward displacement of the supporting platform, while maintaining the feet in place. The following parameters for platform perturbations were used: amplitude of 12 cm, peak velocity of 15 cm/s, and peak acceleration of 100 cm/s$^2$. Perturbations were preceded by randomly varied preparatory intervals, ranging 1–3 s, to prevent anticipation.

## Data analysis

The duration of FoG was calculated in the frequency domain based on the method proposed by Moore et al. (*Moore et al., 2013*; *Moore et al., 2008*), in which the frequency band of the acceleration during normal gait (0–3 Hz) and freezing (3–8 Hz) may be distinguished during walking through changes in frequency power (*Figure 2A,B,C*). Sampling windows of 7.5 s of the lumbar acceleration were defined for the spectrum analysis. In each sampling window, the peak power was calculated for the FoG band (3–8 Hz) and divided by the peak power of the normal gait (0.5–3 Hz). This metric is referred to as the index of FoG. Values above the clinical threshold of 2 were considerate as FoG events (*Moore et al., 2013*; *Moore et al., 2008*).

For analysis of the stepping task, onset of the APA was defined as the time between the abrupt increase of the mediolateral displacement, defined as two standard deviations above the mean of the baseline force, and the onset of the step, defined as two standard deviations above the mean of the baseline foot displacement in the anteroposterior direction (*Figure 2*). Based on these measurements, the time of APA was calculated as the time between the onset of APA and the onset of the step. Mediolateral force amplitude during the step task was normalized by the individual foot size (unit in N/cm).

For analysis of reactive postural control during perturbation trials, root mean square of anteroposterior displacement of the center of mass ($COM_{ap}$) and center of pressure ($COP_{ap}$) were assessed. To calculate $COM_{ap}$, spherical reflective markers were attached at the following anatomical points on the right hemibody: fifth metatarsophalangeal joint, lateral malleolus, lateral knee joint center, greater trochanter and the approximate axis of shoulder rotation (*Figure 2DE*).

Data were extracted and processed with MATLAB (Mathworks, Natick, MA - RRID:SCR_001622) routines, following preliminary visual inspection of signals for individual trials. Data sampling frequency was set at 200 Hz for kinematics and ground reaction forces. Kinematic and ground reaction force data were digitally low-pass filtered with a cut-off frequency of 10 Hz. Signals were filtered through a dual-pass fourth-order Butterworth filter. Estimation of $CoM_{ap}$ displacement was based on the anthropometric model proposed by Winter (*Winter, 1991*), assuming symmetric displacement between two body sides.

## Statistical analysis

Analysis was performed on trial averages of each subject and each stimulation condition (OFF, 60 Hz, 300 Hz) for the four subjects. The following variables were considered: gait time, FoG time, duration and amplitude of APA. We then calculated the percentage of change in the ON (60 Hz and 300 Hz) vs. OFF stimulation conditions using the following calculation: %Change = (*ON (60 or 300 Hz)-OFF)/OFF*.

Improvement was referred as positive values of % change (decrease in gait time, shortening of FoG time, increasing of the amplitude of APA and decreasing of APA duration). Negative values were considered as performance decline. Group averages of percent change for each variable were compared to the basal condition (OFF, no gain) using Student's *t* test. *p*-value set at $\leq 0.05$ for each ON condition (60 and 300 Hz). Analysis was performed using the software Statistica.

## Acknowledgements

This study was funded by the Division of Functional Neurosurgery, Institute of Psychiatry, Hospital das Clinicas, University of São Paulo Medical School; the Brazilian Council of Science and Technology (CNPq, grant number 302628); and the São Paulo Research Foundation (FAPESP, grant number 20265–3).

## Additional information

### Funding

| Funder | Grant reference number | Author |
|---|---|---|
| Conselho Nacional de Desenvolvimento Científico e Tecnológico | 302628 | Luis Augusto Teixeira |

| Fundação de Amparo à Pesquisa do Estado de São Paulo | 20265–3 | Luis Augusto Teixeira |
|---|---|---|
| University of São Paolo | | Erich Talamoni Fonoff |

The funders had no role in study design, data collection and interpretation, or the decision to submit the work for publication.

### Author contributions
Andrea Cristina de Lima-Pardini, Conceptualization, Data curation, Validation, Investigation, Visualization, Methodology, Writing—original draft, Writing—review and editing; Daniel Boari Coelho, Conceptualization, Data curation, Software, Formal analysis, Methodology, Writing—original draft, Writing—review and editing; Carolina Pinto Souza, Conceptualization, Data curation, Investigation, Writing—original draft, Writing—review and editing; Carolina Oliveira Souza, Maria Gabriela dos Santos Ghilardi, Tiago Garcia, Matija Milosevic, Conceptualization, Methodology, Writing—review and editing; Mariana Voos, Methodology, Writing—review and editing; Clement Hamani, Supervision, Writing—review and editing; Luis Augusto Teixeira, Resources, Supervision, Funding acquisition, Writing—review and editing; Erich Talamoni Fonoff, Conceptualization, Resources, Supervision, Funding acquisition, Project administration, Writing—review and editing

### Author ORCIDs
Daniel Boari Coelho http://orcid.org/0000-0001-8758-6507

### Ethics
Clinical trial registration NCT02388204
Human subjects: The study was approved by the department review board and in the ethics committee at Hospital das Clinicas of University of São Paulo and registered in the national clinical research database (CAPPESQ-HCFMUSP #12690213.0.0000.0068), which requires all participants to be previously instructed about the procedures and to give written informed consent prior to study inclusion. Patients included in the present study participated in the clinical trial and also registered at clinical-trials.gov (#NCT02388204). All experiments were performed in accordance with the Declaration of Helsinki

### Decision letter and Author response
Decision letter https://doi.org/10.7554/eLife.37727.006
Author response https://doi.org/10.7554/eLife.37727.007

## Additional files

### Supplementary files
• Transparent reporting form
DOI: https://doi.org/10.7554/eLife.37727.004

### Data availability
All data generated or analysed during this study are included in the manuscript and supporting files. Source data files have been provided for Figures 1 and 2.

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
