## [Decision Letter]

Thank you for submitting your article "Effects of spinal cord stimulation on postural control in Parkinson's disease patients with freezing of gait" for consideration by *eLife*. Your article has been reviewed by two peer reviewers, and the evaluation has been overseen by a Reviewing Editor and Richard Ivry as the Senior Editor. The reviewers have opted to remain anonymous.

The reviewers have discussed the reviews with one another and the Reviewing Editor has drafted this decision to help you prepare a revised submission.

General Assessment:

The reviewers find the attempt to study the effects of SCS on distinct aspects of postural control very interesting, particularly because freezing of fait is a major concern in Parkinson's Disease patients.

Conclusions:

1) Freezing of gait (FoG) is a major feature of Parkinsonism and although SCS seems to be a promising intervention for FoG, its effect on the particular domains of postural control is not known.

2) SCS led to improvement in duration and amplitude of Anticipatory Postural Adjustment (APA).

3) SCS improved FoG and gait time.

4) SCS did not improve Reactive Postural Control (RPC).

5) APA and RPC are managed by different brain circuits and SCS is more effective on APAs but not on RPC due to its influence on the specific circuits that modulate APAs.

This work builds on a previous report from the authors and subsequently aims to identify the effect of SCS on individual aspects of postural control. The objective methods used for quantification of FoG, APA, and RPC such as wireless accelerometers, force platforms, motion analysis systems, and spectrum analysis add to the overall strengths of the paper.

Essential revisions:

The reviewers have highlighted a number of concerns that must be adequately addressed before the paper can be accepted.

1) The rationale for using SCS has been alluded to previous studies in rodents, primates, and humans, however for the convenience of readers it would be worthwhile to add few sentences in the Introduction, to explain the rationale for using SCS for treating PD gait disorders.

2) Figure 1: Needs clarification if time axis in 1A correspond to the same axis in 1C. Vertical axis in D and E need to be clearly labeled.

3) Figure 2 could be improved. Label panel A) as APA and show the duration of APA in the group means. It looks like the peak of APAs also changed but this was not mentioned. B) Label each plot as Gait, APA, RPC, etc. Explain why, in Figure 2B (top), there is lot of change in FoG time and gait time with the placebo 60 Hz stimulation.

4) Although light detection systems are widely used for the evaluation of movement, the authors fail to describe the specific motion analysis system implemented for APA and perturbation response quantification. The Vicon system has a wide variety of products.

5) References need editorial work. Last reference has two different versions. DBS not improving postural responses should include references from St. George et al.

---

## [Author Response]

Essential revisions:The reviewers have highlighted a number of concerns that must be adequately addressed before the paper can be accepted.1) The rationale for using SCS has been alluded to previous studies in rodents, primates, and humans, however for the convenience of readers it would be worthwhile to add few sentences in the Introduction, to explain the rationale for using SCS for treating PD gait disorders.

We added sentences that clarified for the readers how SCS started to be considered a possible new treatment for FoG in PD, and possible neuromodulation of pathways involving brain regions in control of gait initiation.

2) Figure 1: Needs clarification if time axis in 1A correspond to the same axis in 1C. Vertical axis in D and E need to be clearly labeled.

We added the time legend in Graph 1C, which is the same for A1 (time unit of 2s). Vertical axes were labeled on top of each graph.

3) Figure 2 could be improved. Label panel A) as APA and show the duration of APA in the group means. It looks like the peak of APAs also changed but this was not mentioned. B) Label each plot as Gait, APA, RPC, etc. Explain why, in Figure 2B (top), there is lot of change in FoG time and gait time with the placebo 60 Hz stimulation.

We labelled APA in all graphs of panel A, and showed the direction of APA. The duration of APA (time of APA) of group mean is displayed on panel C. The group mean of amplitude of APA (peak) is also shown on panel C. We made clearer the sentence about the increase in amplitude (peak) of APA in the Results section.

We labelled each plot of Figure 2B as suggested. 60Hz was applied as active placebo in the first report and compared to 300Hz, which showed in average better outcomes in gait measures when compared to OFF condition and 60Hz. However, it seems like there are subjects that also experience some response to lower frequencies. As it is not the primary interest in this report, we decided to refer to it as two different active conditions (60Hz and 300Hz) with similar parestesias.

4) Although light detection systems are widely used for the evaluation of movement, the authors fail to describe the specific motion analysis system implemented for APA and perturbation response quantification. The Vicon system has a wide variety of products.

We added the Vicon System details in the “Data collection section”.

5) References need editorial work. Last reference has two different versions. DBS not improving postural responses should include references from St. George et al.

We corrected this imprecision in reference citation.